# Implications of the Onset of Sweating on the Sweat Lactate Threshold

**DOI:** 10.3390/s23073378

**Published:** 2023-03-23

**Authors:** Yuta Maeda, Hiroki Okawara, Tomonori Sawada, Daisuke Nakashima, Joji Nagahara, Haruki Fujitsuka, Kaito Ikeda, Sosuke Hoshino, Yusuke Kobari, Yoshinori Katsumata, Masaya Nakamura, Takeo Nagura

**Affiliations:** 1Department of Orthopaedic Surgery, Keio University School of Medicine, Shinjuku, Tokyo 160-8582, Japan; 2Department of Cardiology, Keio University School of Medicine, Shinjuku, Tokyo 160-8582, Japan; 3Institute for Integrated Sports Medicine, Keio University School of Medicine, Shinjuku, Tokyo 160-8582, Japan; 4Department of Clinical Biomechanics, Keio University School of Medicine, Shinjuku, Tokyo 160-8582, Japan

**Keywords:** lactate threshold, sweat rate, exercise testing, incremental exercise, sweating, perspiration, body temperature regulation, sports, physiology

## Abstract

The relationship between the onset of sweating (OS) and sweat lactate threshold (sLT) assessed using a novel sweat lactate sensor remains unclear. We aimed to investigate the implications of the OS on the sLT. Forty healthy men performed an incremental cycling test. We monitored the sweat lactate, blood lactate, and local sweating rates to determine the sLT, blood LT (bLT), and OS. We defined participants with the OS during the warm-up just before the incremental test as the early perspiration (EP) group and the others as the regular perspiration (RP) group. Pearson’s correlation coefficient analysis revealed that the OS was poorly correlated with the sLT, particularly in the EP group (EP group, r = 0.12; RP group, r = 0.56). Conversely, even in the EP group, the sLT was strongly correlated with the bLT (r = 0.74); this was also the case in the RP group (r = 0.61). Bland-Altman plots showed no bias between the mean sLT and bLT (mean difference: 19.3 s). Finally, in five cases with a later OS than bLT, the sLT tended to deviate from the bLT (mean difference, 106.8 s). The sLT is a noninvasive and continuous alternative to the bLT, independent of an early OS, although a late OS may negatively affect the sLT.

## 1. Introduction

The visualization of exercise tolerance to optimize daily training is encouraged by athletes and supporters. In particular, monitoring metabolic responses, such as the anaerobic threshold (AT), during exercise enables athletes to evaluate their aerobic capacity in real-time [1,2,3,4], which leads to a defined relative workload intensity [4,5]. Previous studies have investigated the ventilatory threshold (VT), measured using an expiration gas analyzer, and the lactate threshold (LT) as good indicators of AT [3,6]. VT testing is costly and requires expertise; thus, it is not easily accessible to recreational or younger athletes without favorable facilities. LT testing requires frequent blood sampling while interrupting strenuous exercise, meaning it does not reflect usual (continuous) exercise. In addition, contamination with other substances, such as sweat, makes the assessment difficult.

In recent years, wearable sensing technology has been the focus of more precise evaluations of physiological responses in the body. We have developed a method to visualize the lactate dynamics of sweat during exercise in a noninvasive, simple, and real-time manner [7,8]. Furthermore, the sweat LT (sLT), assessed using sweat lactate dynamics, is consistent with the LT calculated from blood samples (bLT) and the VT [7]. However, unlike blood lactate levels, changes in sweat lactate levels may be affected by sweating dynamics.

During exercise, sweating occurs with a rise in body temperature that reflects the increased workload [9,10,11]. Previous reports have shown a negative correlation between the local sweating rate and sweat lactate level during exercise [12,13,14], suggesting that increased sweating dilutes the sweat lactate concentration. However, the relationship between the onset of sweating (OS) and sLT remains unclear. If the relationship between the bLT and sLT is strongly dependent on the OS, LT determination using sweat instead of blood may require careful attention to environmental conditions, such as the temperature and humidity, based on the subject’s condition. We aimed to investigate the relationship between the OS and sLT and the effect of the OS on the relationship between the sLT and bLT during incremental exercise.

## 2. Materials and Methods

### 2.1. Participants

Volunteers were recruited from December 2020 to August 2021, and 40 healthy men aged 18–37 years participated in this study. The exclusion criteria were as follows: (1) the presence of comorbidities such as active cardiopulmonary disease, hypertension, or diabetes within two weeks; (2) elite athletes; (3) smokers and those taking medication or performance-enhancing drugs.

The study protocol was conducted in compliance with the ethical guidelines for medical and health research involving human subjects and was approved by the Ethics Committees of Keio University School of Medicine (Approval No. 20180357). Written informed consent was obtained from the study participants for the participation and publication of the findings before enrollment.

### 2.2. Procedures and Protocols

All experiments were performed at sports facilities under similar conditions (24.8 ± 1.9 °C temperature, 42.0 ± 9.3% relative humidity). Prior to the exercise test, the body weight (kg) and height (cm) of each participant were measured. The body mass index (BMI) and body surface area (BSA) were calculated using the formula used by DuBois (body surface area (m^2^) = 0.007184 × height (cm)^0.725^ × weight (kg)^0.425^) [15]. The participants were instructed to avoid drinking alcohol or caffeine for 12 h [16]. They were also required to be well hydrated and refrain from vigorous exercise for 3 h before the exercise test.

Each participant completed incremental exercise using an electromagnetically braked ergometer (POWER MAX V3 Pro; Konami Sports Co., Ltd., Tokyo, Japan). The ergometer seat was adjusted to a favorable height. Two minutes of rest was set to measure the resting outcomes. Immediately after the warm-up for 4 min with a load of 20 W, the exercise test was started at 50 W, increasing by 25 W every minute. The pedaling cadence range was set to 70–80 revolutions per minute (rpm). Each test was terminated owing to (a) subjective exhaustion or (b) reaching 70 rpm for 10 s.

### 2.3. Measurements during the Exercise Test

The sweat rate and sweat lactate concentration were continuously monitored during incremental exercise. The sweat lactate was measured using a sweat lactate sensor chip (Grace Imaging Inc., Tokyo, Japan), which we developed and applied in several studies (Appendix A) [7,8,17,18]. It is a type of electrochemical sensor that detects the potential generated by the redox reaction between the lactate and lactate oxidase. The special protective membrane structure of the sensor allows the aforementioned reaction to last for 30 to 60 min. As a result, the lactate concentration in sweat can be continuously measured throughout the exercise test. Further detailed information regarding the composition and fabrication of the sensor chip is available in our previous study [7]. This sweat lactate sensor chip connected to a wearable sweat lactate sensor was placed on the upper arm [7]. The installation area was carefully cleaned using an alcohol-free cloth to prevent contamination of the sweat sample. The sensor chip was firmly fixed with tape to prevent detachment from the skin. Real-time sweat lactate values were automatically recorded in a connected application device (Grace Imaging Inc., Tokyo, Japan) via Bluetooth at 1 Hz. The sweat lactate value was quantified as the current value because the chip reacts with sweat lactate and generates an electric current [7,8,18,19]. The sLT was defined as the first significant increase in the lactate concentration in sweat above the baseline based on graphical plots [7,8,18,19].

The sweat rate was measured using a pre-calibrated perspiration meter (SKN-200M; SKINOS Co., Ltd., Ueda, Japan) on the upper arm [17] and a Fitbit Inspire HR (Fitbit Inc., San Francisco, CA, USA) was attached to the left wrist, two-finger widths above the ulnar styloid process, to measure the heart rate. The heart rate at rest was measured once at the onset of the warm-up. The sweat rate was recorded at 1 Hz and expressed in milligrams per square centimeter per minute (mg/cm^2^/min). The baseline of the local sweat rate was calculated as the average of the sweat volume in the rest period. The OS was defined as the first significant increase in sweat rate above the baseline based on graphical plots (Figure 1).

To measure blood lactate concentration, a blood sample was obtained from the auricle at rest and every minute during exercise. The lactate concentration in the blood was immediately measured using a lactate analyzer (Lactate Pro2 LT-1730, ARKRAY, Inc., Kyoto, Japan). The blood lactate values were graphically plotted in millimoles per liter (mmol/L). The bLT was determined using graphical plots [3].

### 2.4. Statistical Analysis

The OS, bLT, and sLT were determined visually by three researchers independently in accordance with previous reports [7,17]. We first defined subjects with the OS during the resting period or warm-up as the early perspiration (EP) group, and those with the OS during incremental exercise as the regular perspiration (RP) group. The relationships between the OS, sLT, and bLT were examined using Pearson’s correlation coefficient and a linear regression analysis. For each combination, a one-sample *t*-test on the difference in time was used to examine the fixed error, and a linear regression analysis on the mean and difference in time was used to rule out the proportional error. Furthermore, to investigate the effect of an early OS on the approximation between the bLT and sLT, we compared mean bLT–sLT (s) values between the EP and RP groups using an independent *t*-test. Based on the results of the Kolmogorov–Smirnov test, either an independent *t*-test or Mann–Whitney U test was applied to the age, height, weight, BMI, body water, BSA, room temperature, and relative humidity to compare each variable between the groups. All analyses were performed using IBM SPSS Statistics version 27 (IBM Corp., Armonk, NY, USA), with the statistical significance set at 0.05.

## 3. Results

### 3.1. Participants Characteristics and Physiological Results

All participants completed all procedures and were eligible for the analysis. During exercise, continuous negligible responses were initially detected in the sweat rate, sweat lactate, and blood lactate (Figure 1). Subsequently, they rapidly increased and were defined as the OS, sLT, and bLT, respectively. For all participants, these time points could be clearly defined. Based on the relationship between these points, 17 participants (43%) were categorized into the EP group and 23 participants (57%) into the RP group. Furthermore, in five participants (13%), the OS was later than the bLT.

The descriptive data of the participants in the EP and RP groups are presented in Table 1. No significant differences were observed in any of the characteristics between the groups (*p* > 0.05).

Table 2 shows the physiological results including the load, heart rate, sweat lactate level, and local sweat rate. As in previous reports, the heart rate increased with the exercise intensity and the sweat rate increased after the onset of the incremental exercise. The sweat lactate levels were stable at first and then rapidly increased from the sLT to the end.

### 3.2. Relationship between OS and sLT

Figure 2a shows the relationship between the OS and sLT. A poor correlation was observed in the total cohort (r = 0.38, *p* < 0.05), and no correlation was observed in the EP group (r = 0.12, *p* = 0.61). In contrast, in the RP group, the OS moderately correlated with the sLT (r = 0.56, *p* < 0.05). In the five patients with the OS later than the bLT, the OS showed a strong correlation with the sLT (r = 0.82, *p* = 0.09). The Bland-Altman plot revealed that that the mean difference between the OS and sLT was large in total, especially in the EP group (total, 183.7 s, EP group, 389.6 s, RP group, 85.2 s), which validates the inconsistency between these thresholds (Figure 2b). Figure 2b also shows a positive fixed error (*p* < 0.05, 95% CI: 155.1–274.1) and a proportional error (*p* < 0.05) in total. The fixed error indicated that the sLT occurred after the OS in all the cases.

### 3.3. Relationship between OS and Blood LT

Figure 3a shows the relationship between the OS and bLT. There was no correlation in the total cohort (r = 0.23, *p* = 0.16), EP (r = −0.06, *p* = 0.83) or RP groups (r = 0.30, *p* = 0.16). The Bland-Altman plot revealed that the mean difference between the OS and sLT was large in total, especially in the EP group (total, −195.3 s; EP group, −381.6 s; RP group, −57.6 s), which validates the inconsistency between those thresholds (Figure 3b). Figure 3b also shows both a positive fixed error (*p* < 0.05, 95% CI: 132.1–258.6) and a proportional error (*p* < 0.05) in total. The negative fixed error indicated that the bLT tended to come after the OS, while the bLT preceded the OS in five cases (out of 23 cases, 21%) in the RP group.

### 3.4. Effect of Early OS on the Blood–Sweat Lactate Threshold Approximation

As shown in Figure 4a, the sLT was strongly correlated with the bLT, regardless of early perspiration (total, r = 0.68, *p* < 0.01; EP group, r = 0.74, *p* < 0.01; RP group, r = 0.61, *p* < 0.01). The linear regression analysis revealed that the sLT can be a good indicator of the bLT (total, y = 252.9 + 0.54x, *p* < 0.01, R = 0.68; EP group, y = 217.9 + 0.60x, *p* < 0.01, R = 0.74; RP group, y = 289.1 + 0.49x, *p* < 0.01, R = 0.61). The Bland-Altman plot showed no bias between the bLT and sLT in each group (total, 19.3 s; EP group, 8.0 s; RP group, 27.6 s), and validated a strong agreement between those thresholds (Figure 4b). Figure 4b also shows that in the five cases where the OS occurred later than the bLT (described in blue circle), the sLT was more likely to deviate from the bLT than in other cases. The mean difference between the sLT and bLT among the five was 106.8 s. There was no fixed error (*p* > 0.05, 95% CI: −4.8 to 43.3) or proportional error (*p* > 0.05) in the total group.

Figure 5 also shows that there is no significant difference between the bLT–sLT approximations (=sLT–bLT (s)) in the EP and RP groups (*p* > 0.05).

## 4. Discussion

The most striking finding of this study was the poor association between the OS and sLT during incremental exercises. Moreover, the sLT was strongly correlated with the bLT, independent of the preceding OS. These findings provide further information regarding LT measurements of sweat lactate as a noninvasive, continuous, real-time analytical alternative to blood lactate testing.

The analysis of sweat, which contains various types of physiological information [20], has attracted the attention of researchers and physiologists in the athletic field. In particular, the lactate in sweat is concentrated, and the sLT is reported to correlate with VT and bLT [7], which are common indicators of the aerobic exercise capacity [1,2,3,4,5,6]. However, the relationship between the sweat kinetics and sweat lactate kinetics is yet to be fully investigated; therefore, the interpretation of sweat lactate concentrations remains controversial.

We focused on the effect of an early OS on the time of the sLT by monitoring the local sweating rate and sweat lactate concentration. Herein, no correlation was observed between the OS and sLT in the EP and RP groups. This implies that the sLT is regulated independent of the OS and that the sweat lactate dynamics are not necessarily consistent with the sweat dynamics. Sweating occurs due to an increase in core body temperature [9], and the threshold temperature does not change with exercise [21,22,23] or exercise intensity [11,24,25,26,27]. In contrast, the sLT is observed simultaneously with the bLT [28,29], which manifests as an increased production of the lactate in response to the intensive energy demand in the muscles. Such different mechanisms of OS and sweat lactate generation suggest that the increase in sweat lactate production (indicating sLT) is different from the OS, as shown in this study. We assumed that the rate of increase in exercise intensity used in this study possibly induced a rapid increase in deep body temperature that allowed the OS to occur earlier than the sLT.

Our previous report showed that there was a significant correlation between the bLT and sLT [7]. Herein, we showed that the sLT was strongly associated with the bLT, regardless of the OS. Some studies noted that several physiological changes associated with increased lactate production, such as changes in the autonomic nervous balance, hormones, acid–base equilibrium, and metabolic dynamics, may explain such a simultaneous increase in lactate in different biofluids [7,30]. Environmental and biological factors complicate the interpretation of studies using different subjects because they can induce an earlier OS. Hyperthermal and humid environments induce sweat production [9], and the sweat rate in men is significantly higher than that in women due to the higher sensitivity of the eccrine glands to heat and the differences in the hormonal environment [31,32,33,34]. The high capacity for sweating in well-trained athletes supports heat acclimation to maintain high aerobic tolerance [35,36,37,38]. Thus, the effect of an early OS on the validity of the sLT to estimate the bLT needed to be investigated. As shown in this study, the sLT correlated well with the bLT, even in cases of early OS, which might lead to the greater validity and wider applicability of sweat lactate measurements during exercise. This can be attributed to the irrelevancy of the OS in the sLT due to the difference in the mechanisms of sweating onset and lactate production. In addition, the OS was poorly correlated with the bLT in the EP and RP groups, which suggests that the sLT is superior to the OS as a noninvasive parameter to estimate the bLT.

Insufficient sweating remains a major challenge for the use of lactate sensors. A total of five cases showed a later OS than bLT, and the mean difference between the bLT and sLT (sLT–bLT (s)) was larger than in other cases in the EP and RP groups. This indicates that the sLT is more likely to be delayed than the bLT. Therefore, the device had difficulty in adapting to the sLT measurement under ambient conditions with poor sweating during exercise, such as with low-intensity exercise and a low number of sweat glands due to the genetic background [39]. Adjusting the exercise environment (e.g., humidity and temperature) and duration (e.g., long warm-up and total exercise time) may be required for subjects with delayed perspiration. In addition, the use of hermetic sealing or high local temperatures to promote sweating would overcome insufficient sweating in normal environments. Otherwise, improved sweat lactate sensing devices may be warranted. A sweat sampling patch that operates under novel osmotic extraction principles succeeded in withdrawing sweat without rigorous exertion; however, the data were not continuous [40]. Moreover, further studies are required to examine subjects with less perspiration. In the five cases with delayed OS, the sLT strongly correlated with the OS and tended to be more delayed. Naturally, the sLT was later than the OS because the sLT can be measured only after the sensor detects sweat. Thus, we assumed that the deviation of the sLT from the bLT was attributable to the delay in the OS.

Our findings should be interpreted with the following limitations. First, our results were validated using a single type of exercise. Generally, different exercise protocols require different motor functions and associated metabolic pathways. For example, eccentric exercise leads to lower fatigue and lactate responses than concentric exercise with an equivalent load [41,42]. The OS may be delayed during exercise with a significantly short warm-up time. Second, because of the observational study design, the influence of selection bias cannot be completely excluded. The current study mainly included healthy college-aged men and had a relatively small number of cases, especially untrained participants. Further research is warranted, including with untrained subjects and women, because they may differ in their sweat kinetics, muscle mass, and lactate kinetics. Third, the sweat lactate values were obtained from the upper arms. Regional differences in sweat kinetics during cycling exercises have already been reported, and some have revealed higher sweating rates in the forehead and chest [11,43]. The relationship between the sweat kinetics and sweat lactate kinetics at different sites should be analyzed. Fourth, other parameters of sweat dynamics, such as the local sweat rate, were not examined. The OS is an observed phenomenon at the same deep body temperature, even at different relative intensities of exercise. In contrast, the sweat rate increases significantly with increasing exercise intensity [11]. The effects of different local sweat rates on the sLT should also be examined.

## 5. Conclusions

The results of this study revealed a poor correlation between the OS and sLT, which supports the notion that the sLT is strongly correlated with the bLT, independent of the preceding OS during incremental exercise in healthy men. These findings provide further information regarding LT measurements of sweat lactate as a noninvasive, continuous, real-time analytical alternative to blood lactate testing.

## Figures and Tables

**Figure 1 sensors-23-03378-f001:**
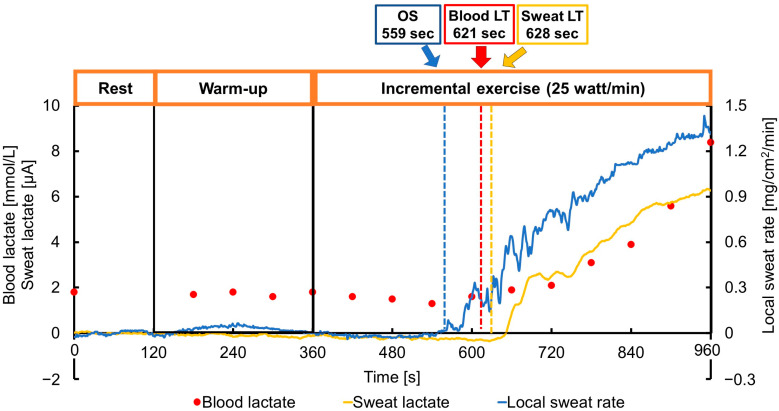
Representative data of the local sweat rate, blood lactate, and sweat lactate throughout the exercise protocol, including the rest and warm-up periods. The onset of sweating (OS), blood lactate threshold (bLT), and sweat lactate threshold (sLT) are indicated by the dotted lines. The OS preceded both the bLT and sLT (35 out of 40 cases, 88%).

**Figure 2 sensors-23-03378-f002:**
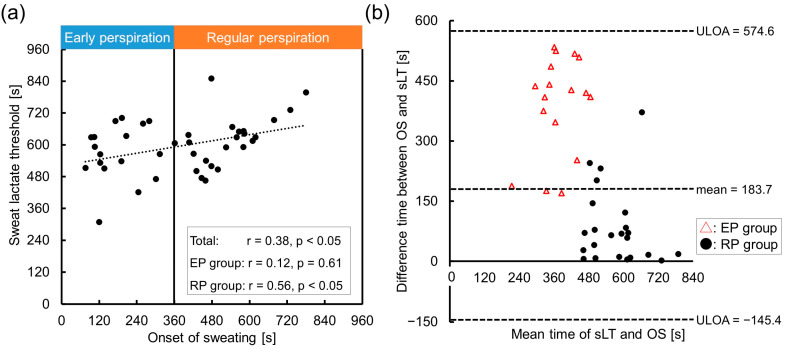
The relationship between the onset of sweating and sweat lactate threshold. (**a**) A scatter plot and approximation line between the onset of sweating (OS) and the sweat lactate threshold (sLT). The correlation was minimal in the total cohort and early perspiration group (EP), while that in the regular perspiration group (RP) was moderate. (**b**) Validity testing using Bland-Altman plots, which indicated the respective differences between the time at the OS and sLT (*y*-axis) against the mean time of OS and sLT (*x*-axis). Red triangles indicate the EP group data and circles indicate the RP group data. Note: r, correlation coefficient; ULOA, upper limit of agreement (=mean + 1.96 × standard deviation); LLOA, lower limit of agreement (=mean − 1.96 × standard deviation).

**Figure 3 sensors-23-03378-f003:**
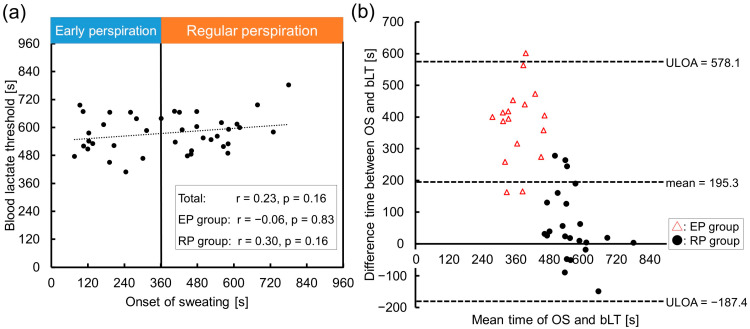
The relationship between the onset of sweating and blood lactate threshold. (**a**) Scatter plot and approximation line between the OS and bLT. No significant correlation was found in the total, early perspiration group (EP), or regular perspiration (RP) groups. (**b**) Validity testing using Bland-Altman plots, which indicated the respective difference between the time at the OS and bLT (*y*-axis) against the mean time of the OS and bLT (*x*-axis). Triangles indicate the EP group data and circles indicate the RP group data. Note: r, correlation coefficient; ULOA, upper limit of agreement (=mean + 1.96 × standard deviation); LLOA, lower limit of agreement (=mean − 1.96 × standard deviation).

**Figure 4 sensors-23-03378-f004:**
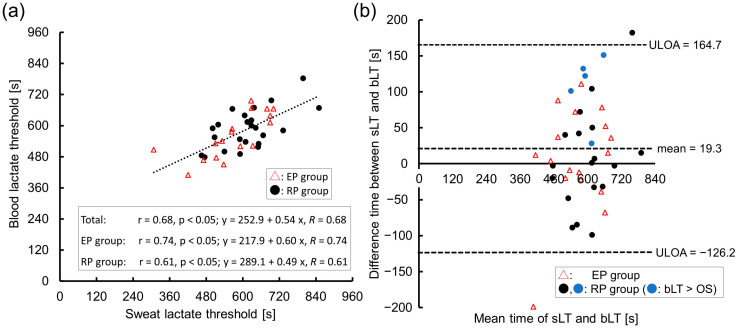
The relationship between the sweat and blood lactate thresholds. (**a**) Scatter plot and approximation line between the sLT and bLT. Significant correlations were found in the total, early perspiration (EP), and regular perspiration (RP) groups. The linear regression analysis also indicated a strong effect of the sLT on the bLT estimation in each group. (**b**) Validity testing using Bland-Altman plots, which indicated the respective difference between the time at the bLT and sLT (*y*-axis) against the mean time at the sLT and bLT (*x*-axis). Red triangles indicate the EP group data, black or blue circles indicate the RP group data. Among the RP group data, blue circles indicate the late perspiration data, where the time of the bLT preceded the time of the OS. Note: r, correlation coefficient; R; multiple correlation coefficient; ULOA, upper limit of agreement (=mean + 1.96 × standard deviation); LLOA, lower limit of agreement (=mean − 1.96 × standard deviation).

**Figure 5 sensors-23-03378-f005:**
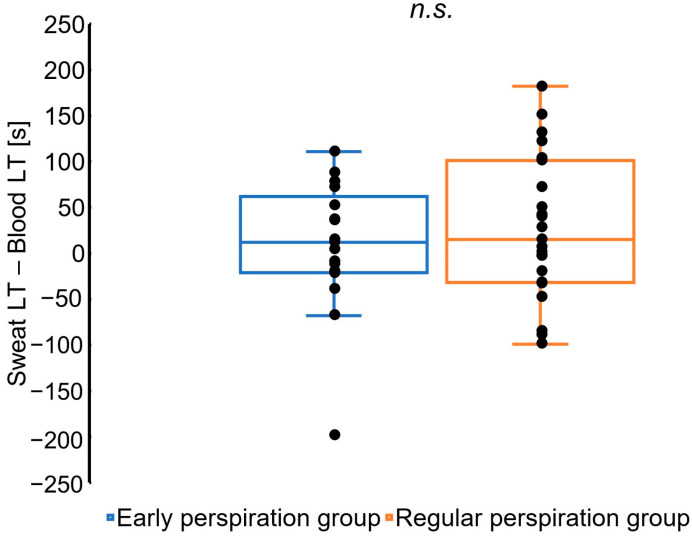
Comparison of the approximations of the sweat lactate threshold to the blood lactate threshold. Box plot of the difference between the times at the blood and sweat lactate thresholds. Black circles indicate the respective data. No significant differences were observed between the early perspiration and regular perspiration groups (*p* = 0.42). Note: n.s., not significance (*p* > 0.05).

**Table 1 sensors-23-03378-t001:** Descriptive data of the participants.

	Total (*n* = 40)	Early Perspiration Group (*n* = 17)	Regular Perspiration Group (*n* = 23)	*p*-Value
Age (years)	21.8 ± 4.0	22.4 ± 4.6	21.4 ± 3.5	0.71
BMI	22.4 ± 2.1	22.7 ± 1.9	22.2 ± 2.3	0.48
BSA (m^2^)	1.8 ± 0.1	1.8 ± 0.1	1.8 ± 0.1	0.86
Body water (%)	57.8 ± 5.4	56.8 ± 5.6	58.6 ± 5.3	0.31
Body fat ratio (%)	17.1 ± 5.0	18.6 ± 5.4	16.0 ± 4.5	0.11
Muscle mass (kg)	53.6 ± 5.8	53.0 ± 5.3	54.0 ± 6.3	0.61
RT (°C)	24.1 ± 1.9	24.2 ± 1.8	24.0 ± 2.0	0.91
RH (%)	42.5 ± 9.3	42.3 ± 8.6	42.6 ± 9.9	0.92

All values are presented as means ± standard deviations or *n* (%). Based on the pre-performed Kolmogorov–Smirnov test, the independent *t*-test or Mann–Whitney U test was applied for intra-group comparisons. BSA, body surface area; RT, room temperature; RH, relative humidity.

**Table 2 sensors-23-03378-t002:** Physiological data of the participants.

	Baseline	At the Warm-Up Onset	At the Incremental Exercise Onset	At the Sweat Lactate Threshold	At the End of Incremental Exercise
Load (watt)	0.0 ± 0.0	20.0 ± 0.0	50.0 ± 0.0	131.7 ± 48.5	261.2 ± 43.6
Heart rate (bpm)	-	79.3 ± 11.1	94.4 ± 14.0	137.0 ± 23.1	172.9 ± 13.8
Sweat lactate (μA)	4.0 ± 1.1	3.9 ± 1.2	3.7 ± 1.2	3.9 ± 1.4	9.6 ± 4.6
Local sweat rate (mg/cm^2^/min)	0.04 ± 0.11	0.01 ± 0.09	0.08 ± 0.17	0.18 ± 0.20	0.84 ± 0.46

The baseline is the average of data in the rest period. All values are presented as means ± standard deviations.

## Data Availability

The datasets supporting the conclusions of this study are available from the corresponding author upon reasonable request.

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
