# Peer review of "Implications of the Onset of Sweating on the Sweat Lactate Threshold"

_sensors, 2023, doi:10.3390/s23073378_

Round 1

Reviewer 1 Report

The authors present a study to study the implications of the sweating onset on sweat lactate threshold. The study is well presented. The detailed comments are:

1. Line 83: What sensor was used? Is it a commercial sensor or developed by the authors? More details should be provided.

2. Figure 1: What sensor was used? Is it a commercial sensor or developed by the authors? More details should be provided.

3. Line 221-223: Based on the presented data, there is no correlation observed between OS and sLT. However, the discussion provides no reasoning as to what the authors hypothesize about why this may be the case. This should be discussed in more detail. 

Author Response

Response to reviewer 1:

The authors present a study to study the implications of the sweating onset on sweat lactate threshold. The study is well presented.

Response: We wish to express our gratitude to reviewer 1 for their insightful and highly valuable comments regarding our manuscript. We believe that the comments have enabled us to significantly improve the manuscript. We have carefully considered these comments and suggestions in the revised version of our manuscript as indicated in our point-by-point responses below:

R1-1&2.  Line 83: What sensor was used? Is it a commercial sensor or developed by the authors? More details should be provided.

Response: Thank you for your comment. We have added a brief explanation of the device structure and how it monitors lactate in sweat. In addition, we have provided the image of the sensor and recording device as a supplementary figure. It is a type of electrochemical sensor that we developed, which detects the potential generated by the redox reaction between lactate and lactate oxidase (Seki Y et al. A novel device for detecting anaerobic threshold using sweat lactate during exercise. Sci Rep. 2021). This device is partially test-launched as a healthcare device and will be launched as a medical device in the near future.

Part of the above explanation has been added to the Materials & Methods section.

Change: “Sweat rate and sweat lactate concentration were continuously monitored during incremental exercise. Sweat lactate was measured using a sweat lactate sensor chip (Grace Imaging Inc., Tokyo, Japan), which we developed and applied in several studies (Supplementary figure 1) [7,8,17,18]. It is a type of electrochemical sensor that detects the potential generated by the redox reaction between lactate and lactate oxidase. The special protective membrane structure of the sensor allows the aforementioned reaction to last for 30 to 60 minutes. As a result, the lactate concentration in sweat can be continuously measured throughout the exercise test. Further detailed information regarding the composition and fabrication of the sensor chip is available in our previous study [7].

(Materials & Methods, Measurements during the exercise test, page 2, lines 83–90)

R1-3. Line 221-223: Based on the presented data, there is no correlation observed between OS and sLT. However, the discussion provides no reasoning as to what the authors hypothesize about why this may be the case. This should be discussed in more detail.

Response: Thank you for your constructive suggestion. We assumed that the poor correlation between OS and sLT was due to the difference in their origin. OS is reportedly induced by high core temperature, while sLT is induced by the rapid increase in sweat lactate production along with high energy demand and intense exercise. Therefore, we hypothesized that the warm-up load (20 watt) and the rate of increase in exercise intensity (25 watt/min) were sufficient to raise participants’ body core temperature to cause OS but insufficient to cause rapid increase in sweat lactate production early. When the exercise intensity is greater, energy demand may increase, leading to an earlier sLT. In such scenario, the relationship between OS and sLT may be different from that in this study.

Part of the above explanation has been added to the Discussion section.

Change: Herein, no correlation was observed between OS and sLT in the EP and RP groups. This implies that sLT is regulated independent of OS and that sweat lactate dynamics are not necessarily consistent with sweat dynamics. Sweating occurs due to an increase in core body temperature [9], and the threshold temperature does not change with exercise [21-23] or exercise intensity [11,24-27]. In contrast, sLT is observed simultaneously with bLT [28,29], which manifests as an increased production of the lactate in response to the intensive energy demand in the muscles. Such different mechanisms of OS and sweat lactate generation suggest that the increase in sweat lactate production (indicating sLT) is different from OS, as shown in this study. We assumed that the rate of increase in exercise intensity used in this study possibly induced a rapid increase in deep body temperature which allowed OS to occur earlier than sLT.

(Discussion, pages 7–8, lines 237–247)

Reviewer 2 Report

The manuscript's scientific interpretation can be widely improved. The facts stated here are interesting and so can be presented with more clarity without causing lesser convolution and stating obvious facts. Also, I recommend citing the following paper:

https://www.mdpi.com/2072-666X/12/12/1513

Author Response

Response to reviewer 2:

The manuscript's scientific interpretation can be widely improved. The facts stated here are interesting and so can be presented with more clarity without causing lesser convolution and stating obvious facts.

Response: We wish to express our gratitude to reviewer 2 for their comments regarding our manuscript after careful review. We believe that the comments have enabled us to significantly improve the manuscript. We have carefully considered your comments and suggestions in the revised version of our manuscript as indicated in our point-by-point responses below:

R2-1. The manuscript's scientific interpretation can be widely improved. The facts stated here are interesting and so can be presented with more clarity without causing lesser convolution and stating obvious facts.

We have carefully revised and clarified parts of the discussion, which may have been convoluted previously. We have also added some reasonings for the facts shown in our study based on previous studies.

Change 1: We have added a reasoning for why there was no correlation observed between OS and sLT.

Herein, no correlation was observed between OS and sLT in the EP and RP groups. This implies that sLT is regulated independent of OS and that sweat lactate dynamics are not necessarily consistent with sweat dynamics. Sweating occurs due to an increase in core body temperature [9], and the threshold temperature does not change with exercise [21-23] or exercise intensity [11,24-27]. In contrast, sLT is observed simultaneously with bLT [28,29], which manifests as an increased production of the lactate in response to the intensive energy demand in the muscles. Such different mechanisms of OS and sweat lactate generation suggest that the increase in sweat lactate production (indicating sLT) is different from OS, as shown in this study. We assumed that the rate of increase in exercise intensity used in this study possibly induced a rapid increase in deep body temperature which allowed OS to occur earlier than sLT.

(Discussion, page 7-8, lines 237–247)

Change 2: We have greatly revised this paragraph and eliminated sentences that was unnecessary to discuss the facts shown in this study.

Our previous report showed that there was a significant correlation between bLT and sLT [7]. Herein, we showed that sLT was strongly associated with bLT, regardless of OS. Some studies noted that several physiological changes associated with increased lactate production, such as changes in autonomic nervous balance, hormones, acid–base equilibrium, and metabolic dynamics, may explain such a simultaneous increase in lactate in different biofluids [7,30]. Environmental and biological factors complicate the interpretation of studies using different subjects because they can induce earlier OS.

(Discussion, page 8, lines 248–254)

Change 3: We have added a reasoning for why strong correlation was observed between sLT and bLT even in cases of early OS.

“As shown in this study, sLT correlated well with bLT, even in cases of early OS, which might provide greater validity and wider applicability of sweat lactate measurement during exercise. This can be attributed to the irrelevancy of OS in sLT due to the difference in the mechanism of sweating onset and lactate production. In addition, OS was poorly correlated with bLT in the EP and RP groups, which suggests that sLT is superior to OS as a noninvasive parameter to estimate bLT.”

(Discussion, page 8, lines 260–265)

Change 4: We have added a reasoning for why there was strong correlation observed between OS and sLT in the five cases with delayed OS.

Moreover, further studies are required to examine subjects with lesser perspiration. In the five cases with delayed OS, sLT strongly correlated with OS and tended to be more delayed. Naturally, sLT was later than OS because sLT can be measured only after the sensor detects sweat. Thus, we assumed that the deviation of sLT from bLT was attributable to the delay in OS.

(Discussion, page 8, lines 278–282)

R2-2. I recommend citing the following paper: https://www.mdpi.com/2072-666X/12/12/1513

Response:

Thank you for your recommendation. We have cited the paper you kindly suggested to broaden our discussion as below.

Change:

“In addition, hermetic sealing or high local temperatures to promote sweating would overcome insufficient sweating in normal environments. Otherwise, improved sweat lactate sensing devices may be warranted. A sweat sampling patch that operates under novel osmotic extraction principles succeeded in withdrawing sweat without rigorous exertion; however, the data was not continuous [40].

(Discussion, page 8, lines 274–278)

Reviewer 3 Report

This manuscript investigated the implications of OS in sLT. Forty healthy males performed an incremental cycling test. Many experimental data were shown to compare the correlation between OS, bLT, sLT, EP, and RP. And some useful results were concluded finally.

The followed issues should be concerned in the minor revision.

1.     In the line 220, sweat rate and sweat lactate concentration was monitored here, while specific data was not shown about sweat rate and lactate concentration, and whether they merely represent the threshold or not. Please denote it.

2.     About warm-up and incremental exercise during cycling, it’s better to measure heart rate and physical consumption of volunteers simultaneously. These two index has great importance in revealing the state of exercise and would help to declare reliability of sLT in fitness.

Author Response

Response to reviewer 3:

This manuscript investigated the implications of OS in sLT. Forty healthy males performed an incremental cycling test. Many experimental data were shown to compare the correlation between OS, bLT, sLT, EP, and RP. And some useful results were concluded finally.

Response: We wish to express our gratitude to reviewer 3 for their insightful and highly valuable comments regarding our manuscript. We believe that the comments have enabled us to significantly improve the manuscript. We have carefully considered these comments and suggestions in the revised version of our manuscript as indicated in our point-by-point responses below:

R3-1. In the line 220, sweat rate and sweat lactate concentration was monitored here, while specific data was not shown about sweat rate and lactate concentration, and whether they merely represent the threshold or not. Please denote it.

Response: Thank you for your comment. As per your suggestion, we have added physiological results to Table 2 and provided a brief explanation of them in the Result section.

Change:

Table 2 shows the physiological results including load, heart rate, sweat lactate level, and local sweat rate. As in previous reports, heart rate increased with the exercise intensity and sweat rate increased after the onset of the incremental exercise. Sweat lactate levels were stable at first and then rapidly increased from the sLT to the end.

(Results, page 4, lines 147–150)

R3-2. About warm-up and incremental exercise during cycling, it’s better to measure heart rate and physical consumption of volunteers simultaneously. These two index has great importance in revealing the state of exercise and would help to declare reliability of sLT in fitness.

Response: We appreciate your suggestion. As you mentioned, we monitored heart rate (HR) as an indicator of the state of participants throughout the exercise test; however, we failed to monitor HR in some cases. We did not measure physical consumption such as expiration gas data. As we mentioned in R3-1, we have added load, heart rate, sweat lactate value, and sweat rate data to Table 2.